# OpenReview forum: "ReAlign: Safety-Aligning Reasoning Models with Verifier-Guided Reinforcement Learning"
_ICLR.cc/2026/Conference — Submitted to ICLR 2026_

### Official Review · Reviewer_DaWQ · 2025-10-18

**Soundness:** 2
**Presentation:** 3
**Contribution:** 3
**Rating:** 4
**Confidence:** 5

**Summary:**

ReAlign tackles a core problem that arises in safety training for LRMs: over-refusal. To mitigate this problem, they propose an RL training approach through a hybrid reward model that incorporates three different reward types focusing on safety, refusal, and helpfulness. They conduct substantial ablations to analyze not only the safety-refusal performance of ReAlign but also the impact of reasoning safety on answer safety, mixtures of thinking vs. non-thinking training/evaluation, and comparison to SFT. While the idea of a hybrid reward model and systematic comparison to SFT training partially overlaps with previous work, the ablations on thinking vs. non-thinking and relationship between reasoning safety and answer safety is quite insightful and helpful in understanding how reasoning impacts safety.

**Strengths:**

1.  While it has been known that reasoning helps with safety, it is unclear how or why it helps. For example, does reasoning help because it explores by pondering on harmful thoughts or because longer safe context drives the model to be more safe? How does reasoning compare to non-reasoning? This paper conducts the following analysis that helps understand these questions.
>- Analysis on reasoning safety vs. answer safety: This work tries to understand how reasoning safety impacts answer safety by directly imposing a reward on the reasoning, which previous work has not done. The results hint that safer reasoning is not necessary for answer safety when training on outcome based rewards and that such RL training approaches might not make the model learn a strong correlation between reasoning and final answer. This suggests that reasoning in the safety domain may be different from reasoning in the math domain.
>- Thinking vs. non-thinking training: As hybrid models with both thinking and non-thinking mode are becoming popular, the results on transferability give insight into how we should compose training data.
2. This paper utilizes online RL for safety. Most work that leverages reasoning for safety adapt SFT/DPO approaches with only a few utilizing RL. More exploration of RL-based approaches are timely.

**Weaknesses:**

1. There is no baseline comparison to prior work. Prior work also tackles the same problem by training LRMs both through SFT [1-3] and RL [4, 5]. Since this work proposes a new methodology, sufficient comparison to baselines is required to see how it fits into previous literature, especially against methods that use RL.
2. Some contributions of this work are a hybrid reward model as a methodology and a comparison to SFT under controlled settings as an analysis. However, a similar idea has been proposed by TARS [5] where they use a dual reward system to solve the problem of safety-only (guard-only) rewards, also conducting controlled experiments comparing to SFT in the same setting as their RL setting. How does ReAlign compare with TARS?
3. [6, 7] also have analysis on reasoning safety vs. answer safety. How the findings of ReAlign are different from [6, 7] should be discussed.
4. Some related work is missing for SFT [1, 3] and RL [5] based approaches. Also, Deliberative Alignment [4] is mentioned as an SFT approach in line 680 but it is an RL approach.
5. I think the claim in lines 439-449 is an overstatement. The reasoning degradation from distilled data (off-distribution) could depend on the teacher model and rejection sampling methodology. Furthermore, SFT data from the target model (on-distribution) not improving safety and maintaining reasoning seems intuitive, even when rejection sampled, because the model does not have much to learn from its own generations.
6. In lines 446-447, it states that reasoning performance was degraded because “ArenaHard-v2 score from 9.5 to 8.4”. However, in lines 310-311, it refers to a similar score difference on the utility benchmarks as “a negligible delta of one another”. The rubric for “negligible” seems arbitrary.


**References**

[1] Wang, Zijun, et al. "Star-1: Safer alignment of reasoning llms with 1k data." arXiv preprint arXiv:2504.01903 (2025).

[2] Jiang, Fengqing, et al. "Safechain: Safety of language models with long chain-of-thought reasoning capabilities." arXiv preprint arXiv:2502.12025 (2025).

[3] Zhang, Yichi, et al. "Realsafe-r1: Safety-aligned deepseek-r1 without compromising reasoning capability." arXiv preprint arXiv:2504.10081 (2025).

[4] Guan, Melody Y., et al. "Deliberative alignment: Reasoning enables safer language models." arXiv preprint arXiv:2412.16339 (2024).

[5] Kim, Taeyoun, et al. "Reasoning as an Adaptive Defense for Safety." arXiv preprint arXiv:2507.00971 (2025).

[6] Zhou, Kaiwen, et al. "The hidden risks of large reasoning models: A safety assessment of r1." arXiv preprint arXiv:2502.12659 (2025).

[7] Zhang, Yichi, et al. "Towards Safe Reasoning in Large Reasoning Models via Corrective Intervention." arXiv preprint arXiv:2509.24393 (2025).

**Questions:**

Additional comments and questions:

1. In line 215, it should be clarified whether the training prompts consist of just harmful prompts or both harmful and benign prompts from WildJailbreak. It would be very interesting if ReAlign worked with only harmless prompts.
2. Same for Appendix D. It would be helpful to clarify what prompt types the x-axis and y-axis for Figure 2 were evaluated on from WildJailbreak.
3. Is no format reward needed?
4. In line 208, could you clarify what “(3) Mix in the remaining data" means?
5. Missing period in line 239.
6. The terminology in-distribution data vs. off-distribution data is a bit confusing as it resembles ID vs. OOD data which refers to a distribution shift from training data. Maybe it could be changed to on-policy data vs. distilled data as it clarifies that the model is different.

---

### Official Review · Reviewer_SBnR · 2025-10-20

**Soundness:** 2
**Presentation:** 3
**Contribution:** 2
**Rating:** 4
**Confidence:** 4

**Summary:**

Large language models, through reinforcement learning on chain-of-thoughts to perform complex tasks, have demonstrated remarkable improvement in reasoning like math or coding problems. However, this performance improvement also comes with safety concerns, since these more capable models are in general also more capable of causing harm. Moreover, prior work has shown that making large reasoning models come with a tax on their reasoning abilities — making them safer also harms their capabilities. This paper studies this problem of how to make reasoning models safe without harming their core capabilities. They introduce a method called ReAlign, where they train a reasoning model with a hybrid reward system, penalizing both unsafe behaviors and refusals, and rewarding safer and high quality responses. Experiments demonstrate that this method is capable of vastly improving the safety of model responses while maintaining core capabilities, giving a potential path to making safe but still capable reasoning models.

**Strengths:**

1. The paper is well-written and structured and therefore easy to follow.

2. I like the use of general reasoning datasets (e.g., AIME 25) in addition to safety-specific benchmarks to showcase the proposed method’s effectiveness. Also, training and evaluation uses two separate judge LLMs, which prevents the risk of reward hacking. This makes the paper’s results more robust and believable.

3. Various ablations regarding the design decisions of the proposed methods are conducted, which I found to be helpful in judging the work.

**Weaknesses:**

The paper has a few key weaknesses (and minor concerns, listed as questions below) in my opinion. I would list them in the order of descending severity, i.e., the first one being most severe and so on.

# Potential Flaw in the Problem Setting (Conflicts with COT monitorability)

The biggest weakness of this paper is actually in the problem setting of attempting to make the chain-of-thoughts themselves safe, instead of just keeping the final response safe. **While the paper studies reward mechanisms not involving safety of the COT itself**, the main method, described in Section 3.1, has a penalty for unsafe COTs. I think this makes the whole method questionable from the COT monitorability perspective.

Below is line 352-353 copied from the paper:

> it is beneficial to include CoT data and to supervise the safety of the CoT.

This conclusion is made based on the single observation that supervising safety of the COT increases safety of the COT. The final answer safety seems unaffected by this. This is a pretty strong statement given prior works that disagree with supervising COT safety directly. Prior works [1, 2] have described how making chain-of-thoughts safer can harm monitorability of modern reasoning LLMs. [1] shows that for a strong reasoning model like o3-mini, prolonged training to apply optimization pressure on the COT directly can lead to obfuscation, where the model can take harmful/unsafe actions while obfuscating their thoughts in the COT, which may make it **much harder** to actually prevent unsafe behaviors. In other words, making COTs safe does not mean that very strong reasoning models like o3-mini will become safe — these models might just learn to obfuscate their chain-of-thoughts. **A very strong counter-argument will be needed supporting why we need to make COTs safe instead of just the final answers, given the results from frontier reasoning models like o3-mini presented in [1], to justify why COTs should be safe as well, given the risk to monitorability that this process may produce.**


# Lack of novelty and contextualization

Several aspects of the paper lack novelty, and do not cite/contextualize the method with prior work. For example, the hybrid reward system is similar to the one in TARS [4], and the data filtering system described below (copied from line 206-208)

> Next, we (2) Filter this data to focus on challenging instances, discarding any prompt where all eight responses were either uniformly safe or uniformly unsafe

is very similar to the filtering mechanism adapted by DAPO [5]. These works are not cited.

# Lack of coverage for experimental results

This is probably the least important weakness, but the experiments in the paper are conducted using a single base model (Qwen3-4B). Running experiments with more base models and RL algorithms, datasets etc. will make the results presented in the paper more robust.

**Questions:**

1. (**Comparison with TARS**) How is the reward system in this paper, combining a safety evaluation reward with general utility/quality reward, fundamentally different from the one proposed in TARS [4]? Both a qualitative comparison describing why this paper’s proposed method would be different/better compared to TARS, and a quantitative comparison showing the benefits of ReAlign over TARS, would significantly improve this paper’s quality.

2. (**Experiment setup for Figure 2**) While Appendix D records the experiment setup for Figure 2 — I am not sure which subsets of the WildJailBreak dataset are used to measure the safety and refusal rates. I.e., is the entire evaluation dataset used to measure both the safety and refusal rates? This will be questionable, since the only way the model can often be safe on a harmful prompt is by refusing to answer it, and so safety and refusal rate has to have a positive correlation. On the other hand, if the authors used the benign prompts to calculate refusal rate and harmful prompts to calculate the safety rate, that would be the correct approach — but this should be properly clarified in the main text of the paper in section 2.1.

3. (**Choice of RL algorithm**) While interesting, why did the authors choose GSPO instead of its more commonly used counterparts like GRPO/RLOO?

4. (**Ablation over reward design**) If one chooses different penalty values for safety vs refusal in equation (1), how does the safety-refusal tradeoff of the resulting model vary?

5. (**Training data construction**) Why are the authors generating 8 rollouts from the base model on a certain subset (it would be great if the authors can clarify how this subset is chosen) using the base model to create the training set of 13.7K COT and 6.7K without COT samples? For online RL, shouldn’t these responses be generated on-policy, using the current model, instead of the base model?


# References

[1] Monitoring Reasoning Models for Misbehavior and the Risks of Promoting Obfuscation, https://cdn.openai.com/pdf/34f2ada6-870f-4c26-9790-fd8def56387f/CoT_Monitoring.pdf

[2] Chain of Thought Monitorability: A New and Fragile Opportunity for AI Safety, https://arxiv.org/abs/2507.11473

[3] Backtracking Improves Generation Safety, https://arxiv.org/abs/2409.14586

[4] Reasoning as an Adaptive Defense for Safety, https://arxiv.org/abs/2507.00971

[5] DAPO: An Open-Source LLM Reinforcement Learning System at Scale, https://arxiv.org/abs/2503.14476

---

### Official Review · Reviewer_z81L · 2025-10-29

**Soundness:** 3
**Presentation:** 3
**Contribution:** 2
**Rating:** 4
**Confidence:** 4

**Summary:**

This paper introduces ReAlign, an RL-based framework for safety alignment of Large Reasoning Models (LRMs) using a Hybrid Reward System that balances safety, helpfulness, and refusal rate. The authors show strong improvements in safety metrics while preserving reasoning capability on several benchmarks.

**Strengths:**

- The overall framework is well-structured and technically solid.


- The proposed method is straightforward to implement while delivering strong empirical performance.


- The results consistently show improved safety without notable degradation in reasoning capabilities.

**Weaknesses:**

### Section 2 analysis lacks novelty.
The analysis presented in Section 2 largely summarizes well-known trends (e.g., the safety–capability–refusal trade-off, the “Safety Tax” phenomenon). While these are correctly described, they do not provide new insights or original empirical findings.

---

### Limited methodological novelty.
Although the framework works well and is well-engineered, its conceptual novelty is limited. The main idea—combining safety, helpfulness, and refusal signals in a hybrid reward—is intuitive and incremental.

---

### Dependence on a single safety verifier.
The reward function relies solely on Qwen3Guard-4B for safety assessment. There is no ablation or comparison using alternative safety verifiers, which raises concerns about overfitting to a specific evaluator.

---

### Insufficient diversity in evaluation datasets.
Safety is evaluated only on the WildJailbreak dataset. A more convincing evaluation should include multiple safety datasets (e.g., HarmBench, JailbreakBench, or custom adversarial prompts). Furthermore, it is unclear on which dataset the refusal rate was computed.

---

### Limited backbone and model scale.
All experiments are conducted on a single backbone (Qwen3-4B).
This weakens the generality of the claims — the authors should include results for different model sizes (e.g., R1-1.5B, R1-7B, R1-32B) and possibly other architectures to show scalability and robustness.

---

### Absence of Related Work
The paper entirely lacks a Related Work section in the main text, which is a serious omission. Furthermore, over the past several months, there has been a surge of research on LRM safety [1,2,3,4,5]. However, the paper provides no positioning, comparison, or discussion relative to these recent works.


[1] STAR-1: Safer Alignment of Reasoning LLMs with 1K Data

[2] The Hidden Risks of Large Reasoning Models: A Safety Assessment of R1

[3] How Should We Enhance the Safety of Large Reasoning Models: An Empirical Study

[4] RealSafe-R1: Safety-Aligned DeepSeek-R1 without Compromising Reasoning Capability

[5] R1-ACT: Efficient Reasoning Model Safety Alignment by Activating Safety Knowledge

**Questions:**

See the Weaknesses.

---

### Official Review · Reviewer_oD9g · 2025-11-01

**Soundness:** 2
**Presentation:** 2
**Contribution:** 1
**Rating:** 2
**Confidence:** 5

**Summary:**

This paper proposes ReAlign, a reinforcement learning framework for safety alignment of LRMs. The method employs a hybrid reward system combining safety verification from a guard model, refusal detection, and helpfulness scoring. The authors apply their approach to Qwen3-4B and demonstrate improved safety rates while maintaining model capabilities across benchmarks like Arena-Hard-V2, AIME-25, and GPQA. The paper also investigates the relationship between Chain-of-Thought safety and final answer safety, and compares RL-based alignment against SFT-based approaches.

**Strengths:**

- The paper proposes a hybrid reward mechanism that attempts to balance safety, helpfulness, and refusal behavior in reasoning model alignment.

- Experimental evaluation across multiple dimensions, including safety, refusal rates, and reasoning capabilities.

**Weaknesses:**

- First, I must be direct: this work seems to lack enough contributions. Authors spend one full section (Sec 2) belaboring an obvious point that there exists a trade-off between safety and refusal rates, or between safety and capability. This is neither surprising nor novel. Everyone working in alignment knows this. The "evidence" it presents (Figure 2, Table 1) merely confirms what is already widely understood in the community. More critically, its actual method occupies less than one page (Sec 3). The so-called "hybrid reward system" in Eq (1) is nothing more than a trivial piecewise function: assign -10 if unsafe, -5 if refusal, otherwise use a standard reward model. This is not a sophisticated design; it is a straightforward heuristic that any practitioner could devise in minutes. There is no principled derivation, no theoretical justification, and no insight into why these specific penalties should work.

- Second, this paper's claim about SFT's poor generalization (Sec 4.5) is not a novel finding. Prior work has already demonstrated that SFT fails to generalize for safety alignment tasks. This has been thoroughly documented in existing literature [1], making its extensive analysis of SFT failures redundant rather than insightful. Authors are actually rediscovering known limitations and presenting them as contributions, which I cannot recognize.

- This paper's evaluation may also be insufficient. First, it only tests on Qwen3-4B. How do we know this approach works on other model families or scales? The generalization claim is entirely unsupported. Second, its safety evaluation relies heavily on Qwen3Guard and WildGuard, both of which are themselves imperfect classifiers. It provides no analysis of inter-annotator agreement, false positive rates, or robustness of its safety metrics. Third, where are comparisons against published safety alignment methods like Constitutional AI, Safe-RLHF, or  STAIR?

- Moreover, this paper does not ablate the specific penalty values (-10 for unsafe, -5 for refusal). Why these numbers? How sensitive are the results to these choices? It does not compare its hybrid reward against simpler alternatives (e.g., binary safe/unsafe rewards with different magnitudes). Table 3 provides some ablation on CoT constraints, but it does not systematically ablate other design choices like the data filtering strategy, group size for advantage computation, or the relative weighting between safety and helpfulness.

- Last but not least, the manuscript exhibits formatting issues that suggest insufficient attention to presentation standards like Table 1 exceeding the page margins.

[1] SFT Memorizes, RL Generalizes: A Comparative Study of Foundation Model Post-training

**Questions:**

The questions are listed in the weakness and this paper may need massive revision.

---

### Meta-Review · Area_Chair_MwP7 · 2026-01-06

**Summary:**

The reviewers raised various significant concerns regarding novelty, motivation and limited evaluation which the authors do not provide any response to. I will be recommending a reject for this submission.

**Reviewer Concerns:**

The main concerns raised by the reviewers were the following:

- Lack of principled design of the reward heuristic.
- Lot of findings of this paper such as the safety vs reasoning capability tradeoff or the superiority of RL over SFT are well known in literature, and thus need to be positioned appropriately.
- Evaluation on a single dataset and model (family) with heavy dependence on specific safety/refusal verifiers without any handling of the fact that the safety/refusal verifiers themselves are imperfect models.
- The hybrid reward system considered in this submission lack appropriate positioning against existing reward and filtering systems.

**Reviewer Scores:**

Given that the authors did not provide a response, I do not think the reviewers would modify their scores significantly.

---

### Decision · Program_Chairs · 2026-01-26

Reject